# Inverted L-Shaped CP Patch Antenna with Corner-Truncated Partial Ground Plane Diagonally Adjoined with Square Branch for L-Band Applications

**DOI:** 10.3390/s21041085

**Published:** 2021-02-05

**Authors:** Phanuphong Boontamchauy, Titipong Lertwiriyaprapa, Chuwong Phongcharoenpanich

**Affiliations:** 1Civil Aviation Training Center, Bangkok 10900, Thailand; phanuphong@catc.or.th; 2Research Center of Innovation Digital and Electromagnetic Technology, Department of Teacher Training in Electrical Engineering, Faculty of Technical Education, King Mongkut’s University of Technology North Bangkok, Bangkok 10800, Thailand; titipong.l@fte.kmutnb.ac.th; 3School of Engineering, King Mongkut’s Institute of Technology Ladkrabang, Bangkok 10520, Thailand

**Keywords:** circular polarization, inverted L-shaped, L-band, patch antenna, square branch

## Abstract

This research proposes an inverted L-shaped patch antenna with a corner-truncated partial ground plane diagonally adjoined to a square branch for L-band applications. The adjoining square branch was used to perturb linear polarization for circular polarization, and the corner-truncated partial ground plane was utilized to enhance the axial ratio bandwidth (ARBW). Simulations were performed, an antenna prototype was fabricated, and experiments were carried out. The simulation and measured results were in good agreement. The proposed antenna could achieve an ARBW of 77.87% (1.09–2.48 GHz). The novelty of this research lies in the concurrent use of a square branch and a corner-truncated partial ground plane to realize wide ARBW in an L-band, rendering the technology suitable for satellite communication and navigation applications.

## 1. Introduction

With the exponential growth in smart wireless devices, compact and low-cost microstrip antennas, especially circularly polarized (CP) patch antennas, are increasingly being deployed in a wide range of devices and applications, such as radio frequency identification (RFID), near-field communication (NFC), wireless local area network (WLAN), and global navigation satellite system (GNSS).

Antenna polarization indicates the direction of electric field vectors in the time domain at a fixed position in space. There are two types of polarization: linear polarization and circular polarization [1]. Circular polarization consists of two linear components of electric field that are perpendicular to each other and equal in magnitude, but have a 90° phase difference. For circular polarization, the electric field rotates in a circle around the direction of propagation [1].

Right-hand circular polarization (RHCP) antennas (clockwise rotation) are commonly used in modern wireless devices. The extent of circular polarization is determined by the axial ratio (AR), which is the proportion of the maximum and minimum semi-axes of the orthogonal electric field components. A perfect CP has an AR of 0 dB; however, in reality an AR ≤ 3 dB is technically acceptable. 

Attempts have been made to realize circular polarization using patch antennas. The dual-fed circularly polarized patch antenna (DCPA) with an external polarizer could achieve near-perfect circular polarization [2]. In DCPA, there are dual feeding points with equal magnitude and a 90° phase difference. In [3], a DCPA with a hybrid coupler achieved a near-perfect CP with a reflection coefficient or return loss bandwidth (RLBW) of 21.76% (1.31–1.63 GHz) and 7.45% ARBW (1.55–1.67 GHz), while a DCPA with a U-slot radiating patch and power divider achieved a 32.08% RLBW (1.091–1.508 GHz) and 21.28% ARBW (1.150–1.424 GHz) [4]. In [5], the Wilkinson network-fed quasi-cross-shaped coupling slot antenna realized a 21.2% RLBW (2.82–3.49 GHz) and 17.43% ARBW (2.88–3.43 GHz), while a DCPA with multiple feeding probes could achieve a 31.6% RLBW (1.2–1.65 GHz) and 20.83% ARBW (1.29–1.59 GHz) [6].

Despite the near-perfect circular polarization, DCPA suffers from fabrication challenges and structural complexity. As a result, single-fed circularly polarized patch antennas (SCPA) are alternatively adopted to realize circular polarization without an external polarizer. Unlike DCPA, the SCPA is straightforward and easy to fabricate by manipulating the feed line, patch shape, ground slot, and/or partial ground plane. In the realization of SCPA, a single-fed patch antenna is manipulated to perturb the LP field to generate a CP field. 

In [7], LP perturbation was realized by the truncation of the single-fed patch. In [8], slits were cut into the circular patch to perturb LP for CP generation, and the proposed method could achieve a 4.92% RLBW (1.011–1.062 GHz) and 2.40% ARBW (1.028–1.053 GHz). In [9], a corner-truncated patch with an coaxial probe feed was used to generate CP with a 11.2% RLBW (1.92–2.16 GHz) and 5.85% ARBW (1.99–2.11 GHz), while a patch with a fractal boundary could realize a 12.7% RLBW (2.35–2.67 GHz) and a 2.12% ARBW (2.376–2.427 GHz) [10]. These findings reveal that LP perturbation by manipulating the radiating patch can achieve a very narrow ARBW.

To improve the ARBW, a circular patch with multiple vias between the radiator and ground plane was proposed and the scheme could achieve a 19.82% RLBW (2.09–2.55 GHz) and 25.2% ARBW (2.1–2.72 GHz) [11]. A coplanar waveguide (CPW) feed with a slot ground could achieve a 72% RLBW (2.26–4.85 GHz) and 66.48% ARBW (2.44–4.87 GHz) [12]. An inverted L-shaped patch with a cross-slot ground plane for the L-band spectrum could achieve a 34% RLBW (1.3–1.83 GHz) and 19.86% ARBW (1.36–1.66 GHz) [13]. 

An inverted L-shaped structure is commonly used in realizing CP antenna. In [14], an inverted L-shaped microstrip with a via connecting to the ground plane achieved a 58.06% RLBW (4.4–8 GHz) and 51.96% ARBW (4.7–8 GHz). An inverted L-shaped microstrip radiator with two L-shaped branches adjoining the partial ground plane for the L-band spectrum could achieve a 44.9% RLBW (1.14–1.8 GHz) and 36.87% ARBW (1.15–1.67 GHz) [15]. In addition, an inverted L-shaped microstrip radiator with a hook-shaped branch adjoining the partial ground plane realized a 56% RLBW (2.25–4.0 GHz) and 63.61% ARBW (2.38–4.60 GHz) [16]. The findings showed that the concurrent use of an inverted L-shaped microstrip and adjoining branch effectively enhanced the ARBW of SCPA. Specifically, the adjoining branch perturbs the LP field to generate a CP field without an external polarizer.

However, the existing inverted L-shaped strip antennas for L-band applications achieved an unsatisfactory ARBW (19.86% and 36.87%) [13,15]. As a result, this research proposes an inverted L-shaped patch antenna with a corner-truncated partial ground plane diagonally adjoined with a square branch for L-band applications. The inverted L-shaped radiating patch could achieve linear polarization (LP), while the adjoining square branch was used to perturb the LP field to generate a CP field. The corner-truncated partial ground plane was utilized to enhance the ARBW. Simulations were performed using CST Studio Suite [17], an antenna prototype was fabricated, and experiments were carried out. The proposed antenna scheme could achieve an ARBW of 77.87% (1.09–2.48 GHz), rendering the technology suitable for satellite communication and navigation applications.

## 2. Antenna Design and Development

Figure 1 illustrates the proposed square-branch corner-truncated partial ground plane patch antenna on an FR-4 substrate (TLM140, Thai Laminate Manufacturer, Bangkok, Thailand). The TLM140 was selected because of its availability in the local market. To design the proposed antenna for 1.5 GHz (center of L-band), the relative permittivity (*ε_r_*), loss tangent (*tanδ*), and thickness (*h*) of the substrate were 4.3, 0.015, and 1.6 mm, respectively. It is noted that the relative permittivity was not readily provided in the TLM140 datasheet at the desired frequency [18]. The curve fitting from available data with a power function was introduced to estimate the relative permittivity at 1.5 GHz. The thickness of the substate was chosen to optimize the reflection coefficient and axial ratio. The substrate was inserted between the inverted L-shaped radiating patch and the corner-truncated partial ground plane with a diagonally adjoining square branch. The radiating patch was connected to the 50 Ω SMA feeding point and the square branch was diagonally adjoined with the corner-truncated ground plane. The proposed inverted L-shape patch antenna was 92 × 96 × 1.6 mm (W × L × H) in dimension.

The inverted L-shaped radiating patch generated linear polarization (LP) field and adjoining square branch was used to perturb the LP field to generate a CP field. The corner-truncated partial ground plane was utilized to realize a wide CP frequency band (L-band frequency spectrum). Table 1 tabulates the optimal dimension of the inverted L-shaped patch antenna with a corner-truncated partial ground plane diagonally adjoined with a square branch.

Prior to the antenna development, the lower resonant frequency (*f_L_* in GHz) of the inverted L-shaped planar monopole antenna (radiating patch) on the partial ground plane in vacuum (Figure 2a) was determined using Equation (1), where *W* and *L* are the width and length (in unit of cm) of the rectangular patch of the monopole antenna and *p* is the distance (in unit of cm) between the rectangular patch and the partial ground plane (i.e., probe length) [19].
(1)$$f_{L}=7.2/\left(L+p+\left(W/2\pi \right)\right)\;\mathrm{GHz}.$$

To realize the lower (first) resonant frequency (*f_L_*) close to the center frequency of the L-band, *W*, *L*, and *p* were thus equal to 4, 3, and 1.15 cm, so the calculated *f_L_* was equal to 1.504 GHz. In Figure 2b, an FR-4 substrate was inserted between the radiating patch and the resized partial ground plane. The FR-4 substrate nevertheless shifted the center frequency. As a result, the partial ground plane was resized (i.e., downsized) to restore the center frequency of the L-band. 

Figure 3 compares the CST simulated reflection coefficient or return loss bandwidth (RLBW) of the inverted L-shaped planar monopole antenna on a partial ground plane in vacuum with that on an FR-4 substrate. The resizing of the partial ground plane resulted in two resonant frequencies within the L-band (1–2 GHz). However, the AR of the inverted L-shaped antenna on the resized partial ground plane was greater than 3 dB (>3 dB) in the L-band, as shown in Figure 4. To realize an AR below 3 dB (≤3 dB), a diagonally adjoining branch was integrated into the proposed antenna scheme as part of the partial ground plane.

A transmission-line model was introduced to analyze a microstrip patch antenna using a series and parallel resonant circuit, where the bandwidth is given by Equations (2) and (3), respectively, and the resonant frequency can be calculated using Equation (4) [20].
(2)$$BW=\;\frac{1}{Q}=\;\frac{R}{\omega_{0}L}=2\frac{\Delta \omega }{\omega_{0}},$$
(3)$$BW=\;\frac{1}{Q}=\;\frac{1}{\omega_{0}RC}=2\frac{\Delta \omega }{\omega_{0}},$$
(4)$$\omega_{0}=1/\sqrt{LC}.$$

Figure 5 shows the equivalent circuit model of the inverted L-shaped planar monopole antenna on a partial ground plane in vacuum and with the FR-4 substrate. In the figure, the microstrip patch can be considered as a parallel resonant circuit of inductor (*L_p_*), capacitor (*C_p_*), and finite resistance (*R_p_*) [1]. The corner feeding to the patch can be modeled as a series inductor (*L_s_*) with a capacitor (*C_s_*) and resistance (*R_s_*). In vacuum, the initial values of *R*, *L*, and *C* can be calculated using Equations (2)–(4) for the operating frequency of 1.5 GHz. The equivalent circuit model was subsequently optimized using the Keysight Advanced Design System (ADS) software for optimal component values. 

The introduction of the FR-4 substrate (replacing vacuum with FR-4) and resizing of the partial ground plane altered the optimal component values of the equivalent circuit, as tabulated in Table 2; therefore, double resonance can be obtained. Meanwhile, the wider bandwidth could be achieved by increasing *R_s_* and decreasing *R_p_*. Figure 6 compares the simulated reflection coefficient of an equivalent circuit using ADS in vacuum and with an FR-4 substrate.

Figure 7 shows the evolutionary stages of the inverted L-shaped patch antenna with a branch structure diagonally adjoining the partial ground plane: Antennas I, II, III, and IV. Each of Antenna I–IV was simulated with CST for investigating mainly the CP. In Antenna I, the diagonally adjoining branch is of symmetrical L-shaped type and located at the farthest corner of the ground plane [15,16], as shown in Figure 7a. The adjoining branch was used to perturb the LP field (from the inverted L-shaped patch antenna) to generate a CP field. In Antenna II, the diagonally adjoining branch of the square ring shape was used to excite CP in the upper L-band frequency (Figure 7b). In Antenna III, the adjoining branch of the solid square shape was used to mitigate the coupling effect between the inverted L-shaped radiating patch and the adjoining branch (Figure 7c). In Antenna IV, the corner-truncated partial ground plane was introduced to enhance ARBW, as shown in Figure 7d.

Figure 8 compares the simulated reflection coefficients of Antennas I, II, III and IV and Figure 9 shows the corresponding simulated axial ratio. In Antenna I, the simulated RLBW (|S_11_| ≤ −10 dB) was 50.84% (1.1–1.85 GHz), achieving circular polarization (AR ≤ 3 dB) with a 29.45% ARBW (1.1–1.48 GHz). The simulated RLBW and ARBW of Antenna II were 53.69% (1.09–1.89 GHz) and 64.53% (1.06–2.07 GHz). The square ring branch in Antenna II significantly enhanced the axial ratio bandwidth (64.53% vs. 29.45% for Antenna I).

The simulated RLBW and ARBW of Antenna III were 54.66% (1.09–1.91 GHz) and 64.55% (1.07–2.09 GHz). In Figure 9, the solid square branch of Antenna III rendered AR close to 1 dB in the upper L-band (1.5–2.0 GHz). The low AR (≤1 dB) was attributable to the larger surface area of the solid square branch, vis-à-vis the square ring branch (Antenna II). The larger surface area mitigated the coupling effect between the inverted L-shaped radiating patch and the adjoining branch (Figure 10). In Antenna IV, the simulated RLBW and ARBW were 53.12% (1.106–1.906 GHz) and 70.88% (1.02–2.14 GHz). The triangular corner truncation of the partial ground plane enhanced the ARBW from originally 64.55% (Antenna III) to 70.88%. The enhanced ARBW rendered Antenna IV suitable for a wide range of L-band applications. 

Figure 10a,b illustrate the coupling effect between the inverted L-shaped radiating patch and the adjoining branch at the center frequency of 1.5 GHz and the 0° phase of Antennas II and III. The coupling effect was mitigated as the surface area of the square branch increased (Antenna III).

Figure 11a–d illustrate the surface current distribution of the inverted L-shaped patch antenna with a corner-truncated partial ground plane diagonally adjoined with a square branch (Antenna IV) at the center frequency (1.5 GHz) at 0°, 90°, 180°, and 270° phase. The surface current density was very high around the feed line and relatively high near the adjoining area between the square branch and the corner-truncated partial ground plane. The perturbation induced by the diagonally adjoining square branch generated right-hand circular polarization (RHCP). At 0° phase, the surface current vectors traveled in the +X direction and rotated to the -Y direction at 90° phase. At 180° phase, the surface current vectors moved in the -X direction and turned to the +Y direction at 270° phase. The surface current rotated clockwise, achieving RHCP in the -Z direction.

## 3. Parametric Study

This section investigates the effects of key antenna parameters on the RLBW and ARBW of the inverted L-shaped patch antenna with a corner-truncated partial ground plane diagonally adjoined with a square branch. The key antenna parameters included branch dimension (*b*), corner-truncated size (*t*), vertical length (*L*_1_) and horizontal length (*L*_4_) of the inverted L-shaped radiating patch, and the distance between the radiating patch and the edge of the substrate (*d*). The parametric studies were designed to investigate the possible effects of varying the antenna structures. Therefore, it can provide only proper trends for initiation and tuning the design but cannot be used to visualize the full range of antenna behavior.

### 3.1. The Effect of Branch Dimension (b) 

Figure 12 illustrates the simulated RLBW and ARBW under variable branch dimensions (*b*): 23, 26, and 29 mm. As *b* increased, the RLBW shifted to the left toward lower frequency while the ARBW increased. However, with excessively large *b* (29 mm), the AR became larger than 3 dB (>3 dB), resulting in non-circular polarization. As a result, the optimal *b* was 26 mm.

### 3.2. The Effect of Corner-Truncated Size (t) 

Figure 13 illustrates the simulated RLBW and ARBW under variable corner-truncated size (*t*): 17, 27, and 37 mm. The variation in *t* had minimal effect on the RLBW, while the ARBW increased with the increase in *t*. However, with *t* = 37 mm, AR was mostly greater than 3 dB (>3 dB), resulting in non-circular polarization. As a result, the optimal *t* was 27 mm.

### 3.3. The Effect of Vertical Length of Inverted L-Shaped Radiating Patch (L_1_)

Figure 14 illustrates the simulated RLBW and ARBW under a variable vertical length of inverted L-shaped radiating patch (*L*_1_): 74, 76, and 78 mm. The length of *L*_1_ was approximately equal to the quarter wave length of the lowest operating frequency (1 GHz). As *L*_1_ increased, RLBW shifted to the left toward a lower frequency, while ARBW remained relatively unchanged with the increase in *L*_1_, except in the upper L-band spectrum. In the upper L-band spectrum, AR approached 1 dB as the *L*_1_ increased, resulting in near-perfect circular polarization. The optimal *L*_1_ was thus 76 mm.

### 3.4. The Effect of Horizontal Length of Inverted L-Shaped Radiating Patch (L_4_)

Figure 15 illustrates the simulated RLBW and ARBW under the variable horizontal length of the inverted L-shaped radiating patch (*L*_4_): 24, 29, and 34 mm. As *L*_4_ became excessively large (34 mm), the reflection coefficient (|S_11_|) was greater than −10 dB (|S_11_| > −10 dB), resulting in an impedance mismatch. As *L*_4_ increased (from 29 to 34 mm), AR became smaller but ARBW became narrower. The optimal *L*_4_ was 29 mm.

### 3.5. The Effect of Distance between the Radiating Patch and the Edge of the Substrate (d)

Figure 16 illustrates the simulated RLBW and ARBW under variable distance between the radiating patch and the edge of the substrate (*d*): 14, 19, and 24 mm. As *d* increased, the reflection coefficient (|S_11_|) deteriorated while AR became smaller. The optimal *d* was thus 19 mm. 

## 4. Results and Discussion

A prototype antenna of the inverted L-shaped patch antenna with a corner-truncated partial ground plane diagonally adjoined with a square branch (Antenna IV) was fabricated and experiments were carried out (Figure 17a,b). During the fabrication, the relative permittivity of the actual substrate was characterized using a near-field transmission-line measurement technique implemented by a modified complementary square ring resonator (CSRR) [21]. A 50 Ω transmission line acted as a sensor with the Minkowski fractal pattern of CSRR. The extraction result of FR-4 (TLM140) was the relative permittivity of 4.308 at 1.5 GHz (average of 5 measurements, as shown in Table 3). The accuracy of the method used for characterization was limited by the air gap effect between the material under test (MUT) and CSRR, therefore the extracted values of relative permittivity could be obtained by a number of measurements. In addition, the resonant frequency and magnitude response of CSRR were dependent on the thickness of the substrate. Therefore, substrate thickness must be reasonably chosen to yield the sufficient accuracy of the measured relative permittivity and loss tangent of the material under test (MUT). In the experiment in an anechoic chamber, the antenna prototype on a rotating table was used as the receiving antenna, and a standard CP reference antenna (ETS-Lindgren’s Model 3102 Conical Log Spiral series) was used as the transmitting antenna (Figure 17c). The results were measured by a vector network analyzer (model E5061B, Agilent, Santa Clara, CA, USA). The performance metrics included the reflection coefficient, AR, gain, and radiation pattern. 

Figure 18 compares the simulated and measured reflection coefficients of the proposed antenna. The simulated and measured RLBWs (|S_11_| ≤ −10 dB) were 53.12% (1.106–1.906 GHz) and 62.37% (1.07–2.04 GHz), with the center frequency of 1.55 GHz. Figure 19 compares the simulated and measured AR of the proposed antenna. The simulated and measured ARBWs (AR ≤ 3 dB) were 70.88% (1.02–2.14 GHz) and 77.87% (1.09–2.48 GHz). The simulation and measured results were agreeable. However, there are some discrepancies between the simulated and measured reflection coefficients and axial ratios. The degradation of the measured reflection coefficient in the upper L-band frequency (1.8–2.0 GHz) and the improved measured reflection coefficient in the lower L-band frequency (1.0–1.5 GHz) could be attributed to the soldering of the SMA in the antenna fabrication. The same rationale also affected the simulated and measured axial ratios of the proposed antenna. On the other hand, the simulation was carried out under ideal conditions in the absence of soldering.

Figure 20 compares the simulated and measured gains of the proposed antenna. The simulated and measured maximum gains were 3.0 dBi at 1.7 GHz and 2.76 dBi at 1.6 GHz, respectively. The simulation and measured results were in good agreement. The simulation and measured radiation patterns of the proposed antenna at the center frequency (1.5 GHz) are illustrated in Figure 21a,b. RHCP radiates in the -Z direction (clockwise) and LHCP in the +Z direction (counterclockwise), resulting in bi-directional radiation. The simulation and measured results were in good agreement. Table 4 summarizes the antenna parameters, RLBW, and ARBW of previous studies and this current research. Given the target L-band frequency (1–2 GHz), the ARBW of the proposed antenna scheme (77.87%) was significantly wider than that of [13,15] (19.86% and 36.87%, respectively). In comparison with [16,22,23,24], for C- and S-band applications (30.8–72.9%), the proposed antenna scheme could achieve a wider ARBW (77.87%).

Essentially, the proposed inverted L-shaped patch antenna with a corner-truncated partial ground plane diagonally adjoined with a square branch for L-band spectrum could achieve an ARBW of 77.87% (1.09–2.48 GHz), rendering the technology suitable for satellite communication and navigation applications.

## 5. Conclusions

This research proposed an inverted L-shaped patch antenna with a corner-truncated partial ground plane diagonally adjoined to a square branch for the L-band spectrum (1–2 GHz). The inverted L-shaped radiating patch generated an LP field, and the adjoining square branch was used to perturb the LP field to generate a CP field. The corner-truncated partial ground plane was utilized to enhance the ARBW. Simulations were performed, an antenna prototype was fabricated, and experiments were carried out. The simulated and measured RLBWs (|S_11_| ≤ −10 dB) were 53.12% (1.106–1.906 GHz) and 62.37% (1.07–2.04 GHz), and the corresponding ARBWs were 70.88% (1.02–2.14 GHz) and 77.87% (1.09–2.48 GHz). The simulated and measured maximum gains were 3.0 dBi at 1.7 GHz and 2.76 dBi at 1.6 GHz, and the proposed antenna achieved RHCP in a clockwise direction. Overall, the simulation and measured results were in good agreement. In essence, the proposed inverted L-shaped patch antenna with a corner-truncated partial ground plane diagonally adjoined with a square branch for the L-band spectrum could achieve an ARBW of 77.87% (1.09–2.48 GHz), rendering the technology suitable for satellite communication and navigation applications. Subsequent research will experiment with a stacked patch antenna by incorporating a metasurface structure into the antenna scheme to enhance the maximum gain and utilize the principle of equivalent circuit to investigate the proposed antenna in different evolutionary stages. In addition, the miniaturization of the proposed antenna could be achieved by using a higher-permittivity substrate or a magneto-dielectric metamaterial.

## Figures and Tables

**Figure 1 sensors-21-01085-f001:**
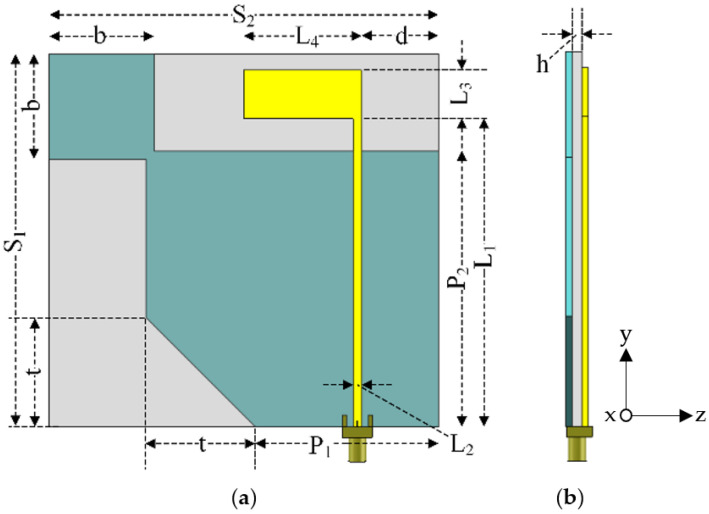
The geometry of the square-branch corner-truncated partial ground plane patch antenna: (**a**) front view, (**b**) side view.

**Figure 2 sensors-21-01085-f002:**
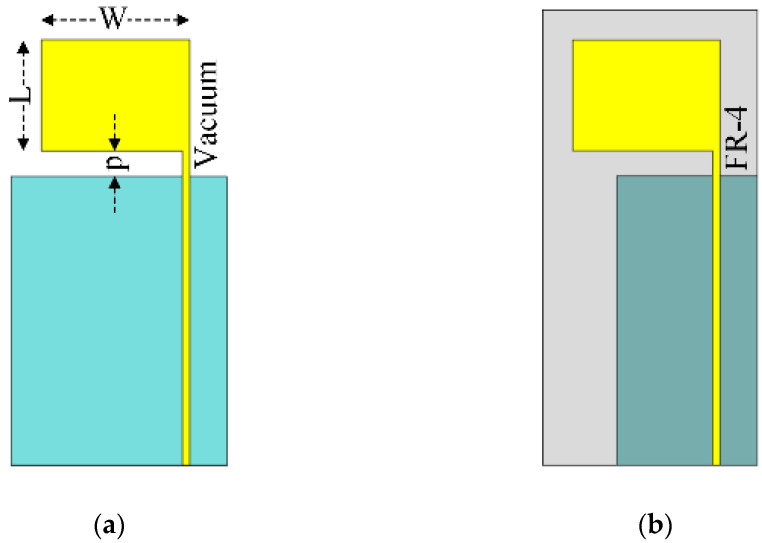
Inverted L-shaped planar monopole antenna on the partial ground plane: (**a**) in vacuum, (**b**) with FR-4 substrate.

**Figure 3 sensors-21-01085-f003:**
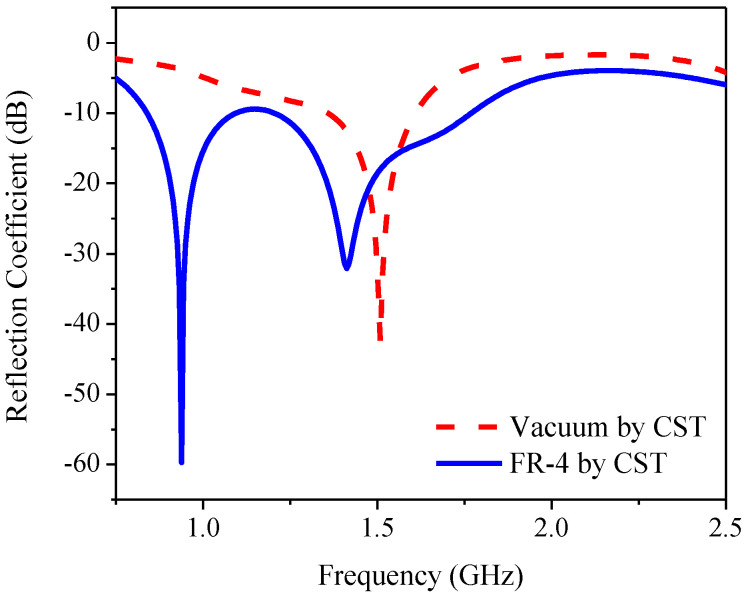
The reflection coefficient of the inverted L-shaped planar monopole antenna on a partial ground plane in vacuum and with FR-4 substrate.

**Figure 4 sensors-21-01085-f004:**
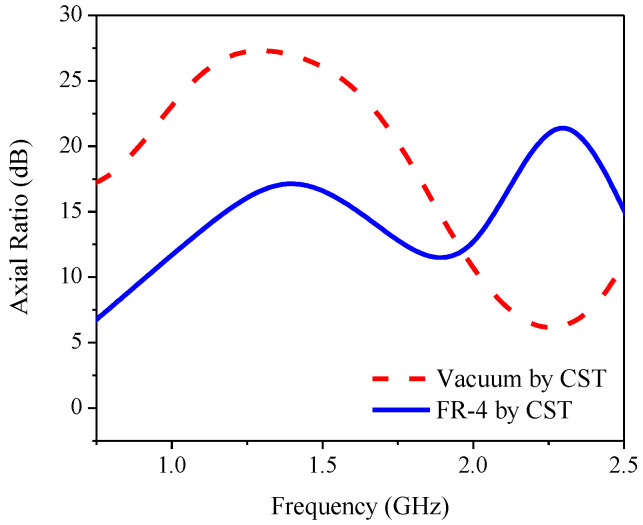
The axial ratio of the inverted L-shaped planar monopole antenna on a partial ground plane in vacuum and with FR-4 substrate.

**Figure 5 sensors-21-01085-f005:**
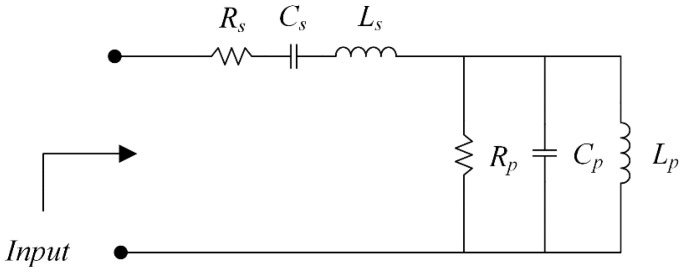
The equivalent circuit model of an inverted L-shaped planar monopole antenna on a partial ground plane in vacuum and with FR-4 substrate.

**Figure 6 sensors-21-01085-f006:**
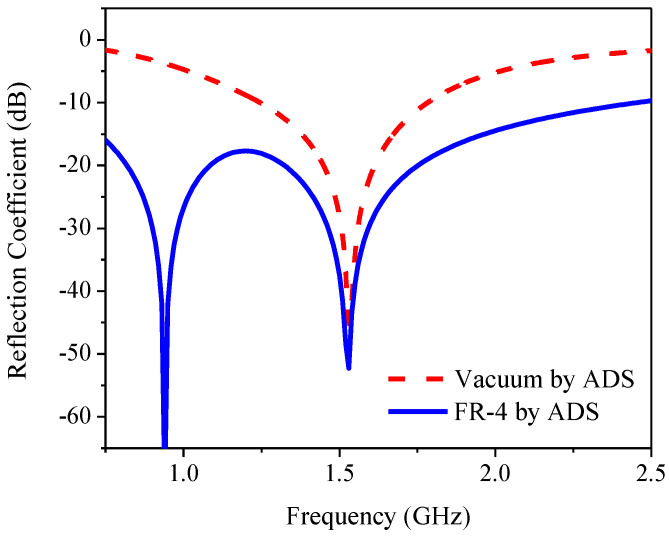
Simulated reflection coefficient of equivalent circuit of inverted L-shaped planar monopole antenna on a partial ground plane in vacuum and with FR-4 substrate.

**Figure 7 sensors-21-01085-f007:**
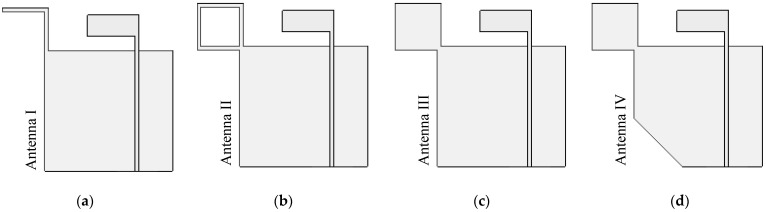
The evolutionary stages of the branch inverted L-shaped partial ground plane patch antenna: (**a**) Antenna I, (**b**) Antenna II, (**c**) Antenna III, (**d**) Antenna IV.

**Figure 8 sensors-21-01085-f008:**
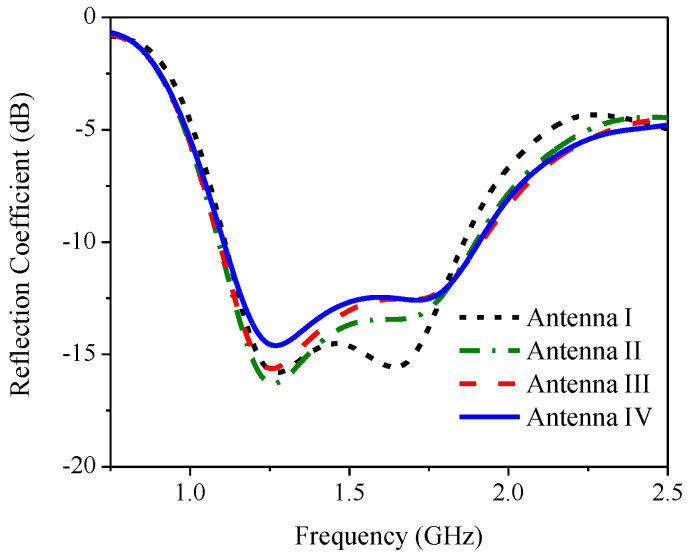
The simulated reflection coefficient of Antennas I, II, III and IV.

**Figure 9 sensors-21-01085-f009:**
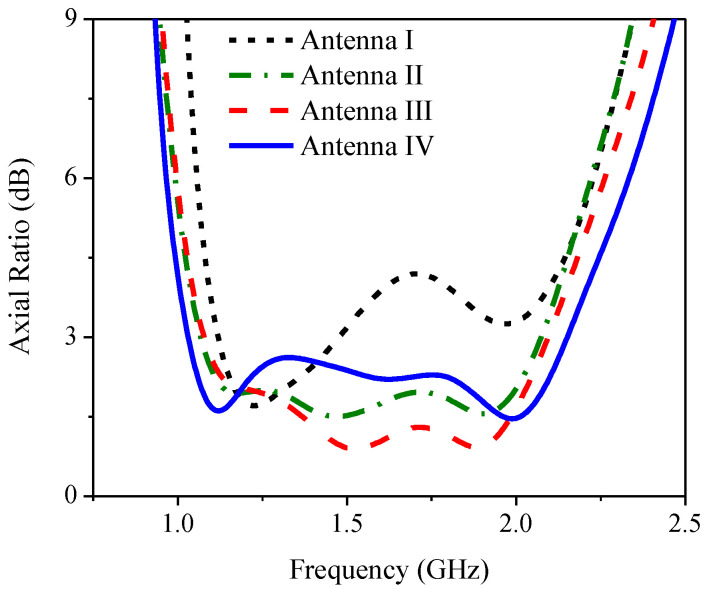
The simulated axial ratio of Antennas I, II, III and IV.

**Figure 10 sensors-21-01085-f010:**
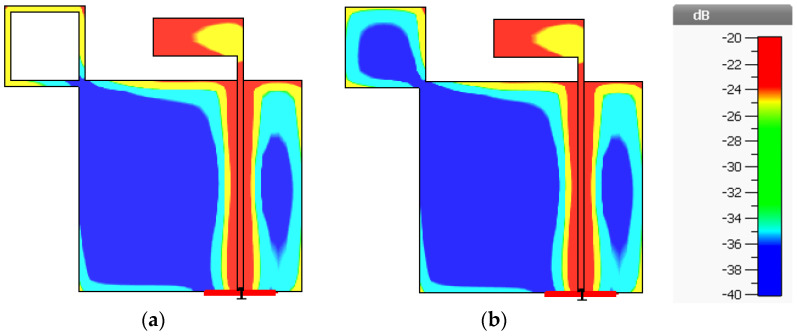
The coupling effect between the inverted L-shaped radiating patch and the adjoining branch at the center frequency of 1.5 GHz and 0° phase: (**a**) Antenna II (square ring branch), (**b**) Antenna III (solid square branch).

**Figure 11 sensors-21-01085-f011:**
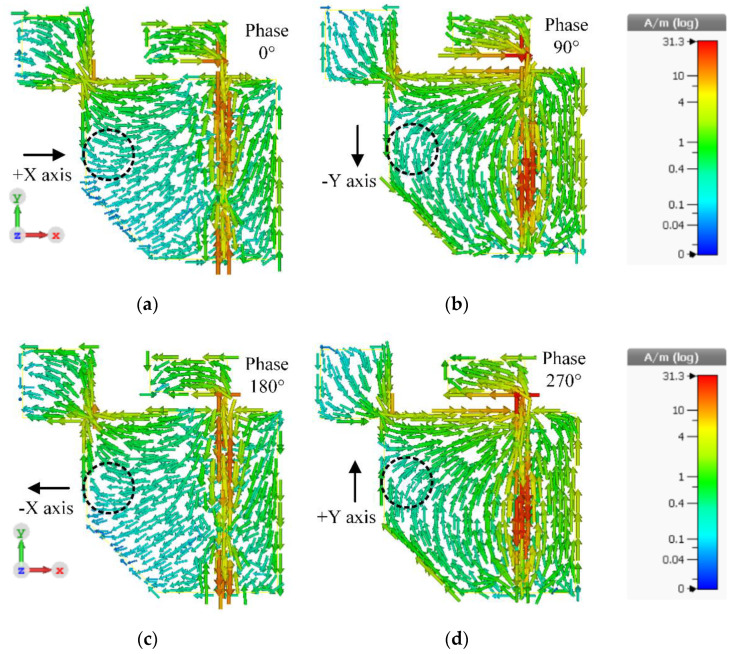
The surface current distribution on the proposed antenna (Antenna IV) at different phases: (**a**) 0°, (**b**) 90°, (**c**) 180°, (**d**) 270°.

**Figure 12 sensors-21-01085-f012:**
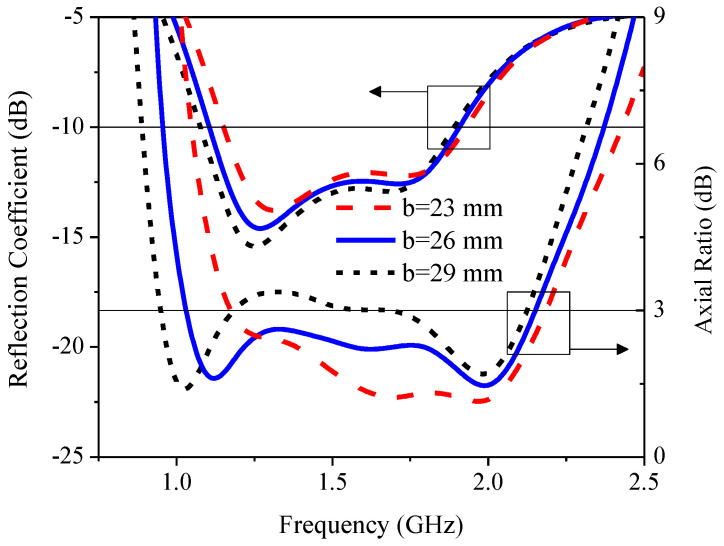
The simulated reflection coefficient and axial ratio under variable branch dimensions (*b*).

**Figure 13 sensors-21-01085-f013:**
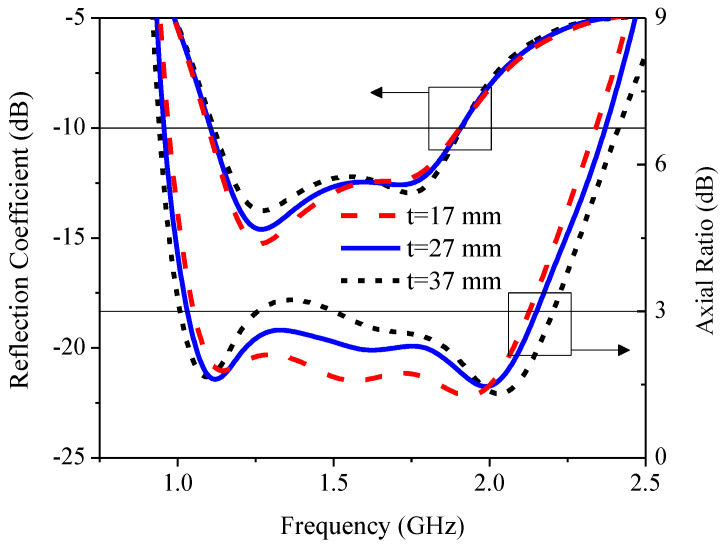
The simulated reflection coefficient and axial ratio under variable corner-truncated size (*t*).

**Figure 14 sensors-21-01085-f014:**
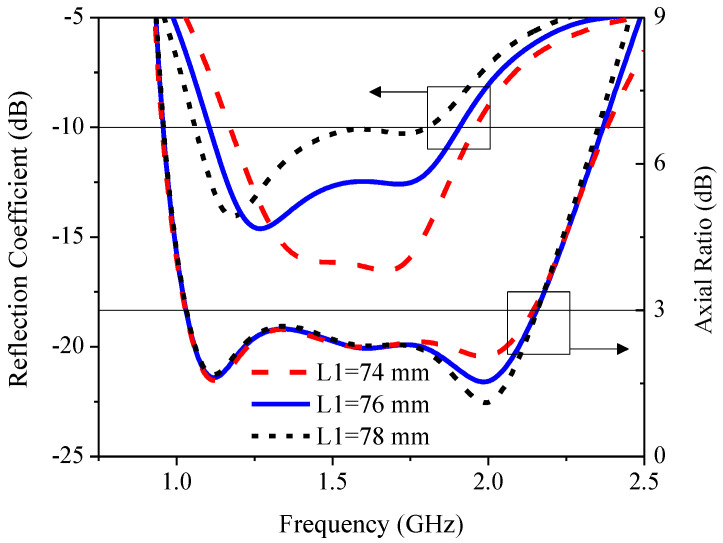
The simulated reflection coefficient and axial ratio under variable vertical length (*L*_1_) of the radiating patch.

**Figure 15 sensors-21-01085-f015:**
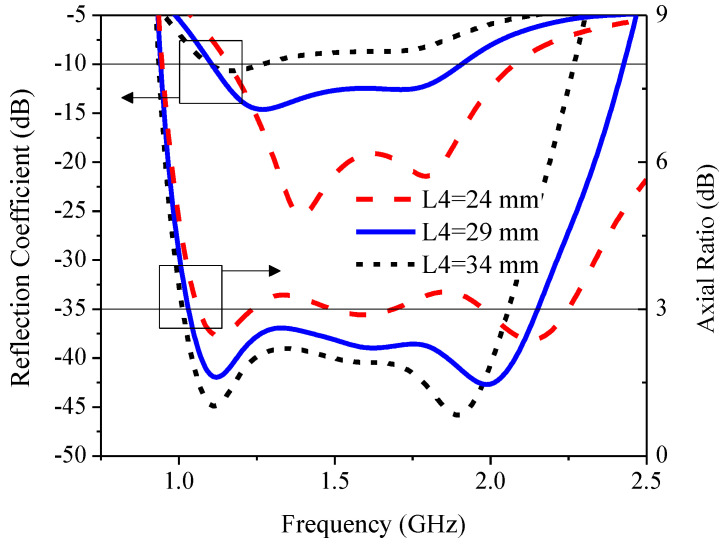
The simulated reflection coefficient and axial ratio under variable horizontal length (*L*_4_) of radiating patch.

**Figure 16 sensors-21-01085-f016:**
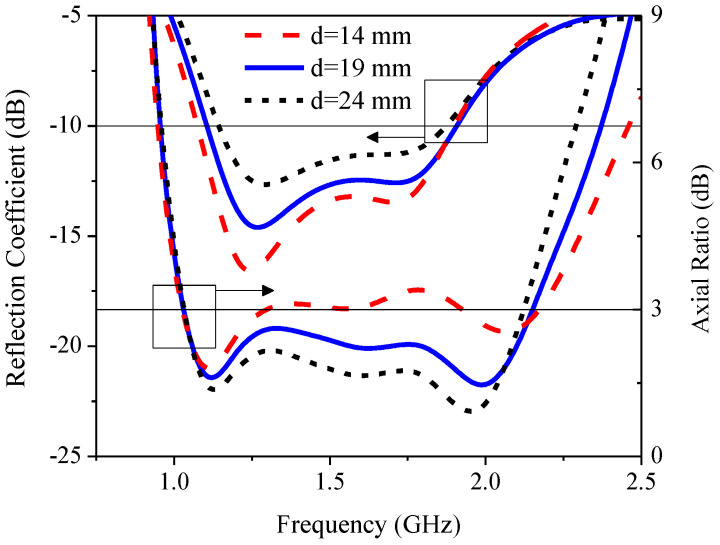
The simulated reflection coefficient and axial ratio under variable distance between the radiating patch and the edge of the substrate (*d*).

**Figure 17 sensors-21-01085-f017:**
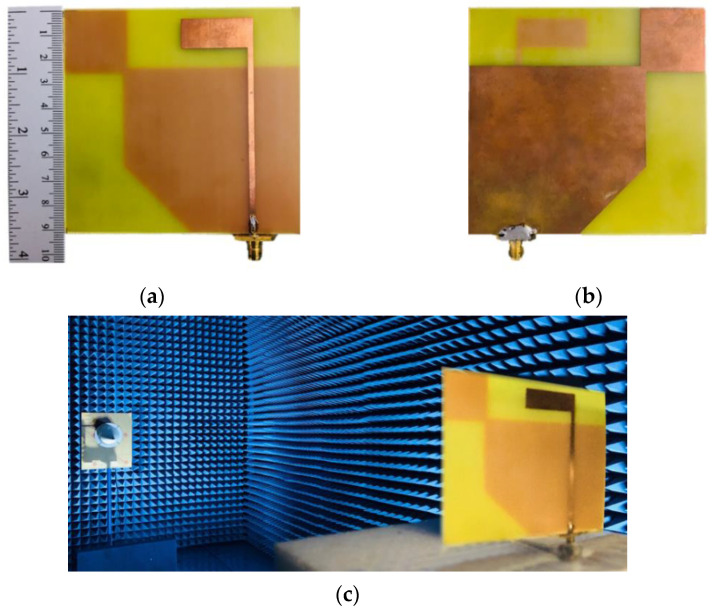
The prototype of the inverted L-shaped patch antenna with a corner-truncated partial ground plane diagonally adjoined with square branch: (**a**) front, (**b**) rear, (**c**) in anechoic chamber.

**Figure 18 sensors-21-01085-f018:**
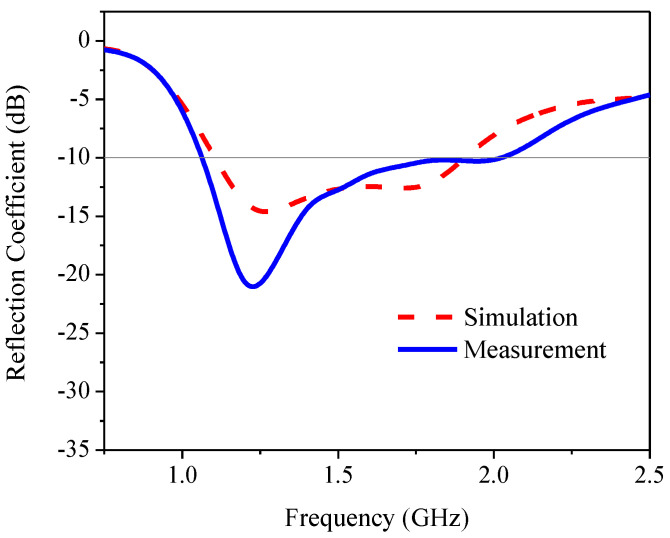
The simulated and measured reflection coefficient of the proposed antenna.

**Figure 19 sensors-21-01085-f019:**
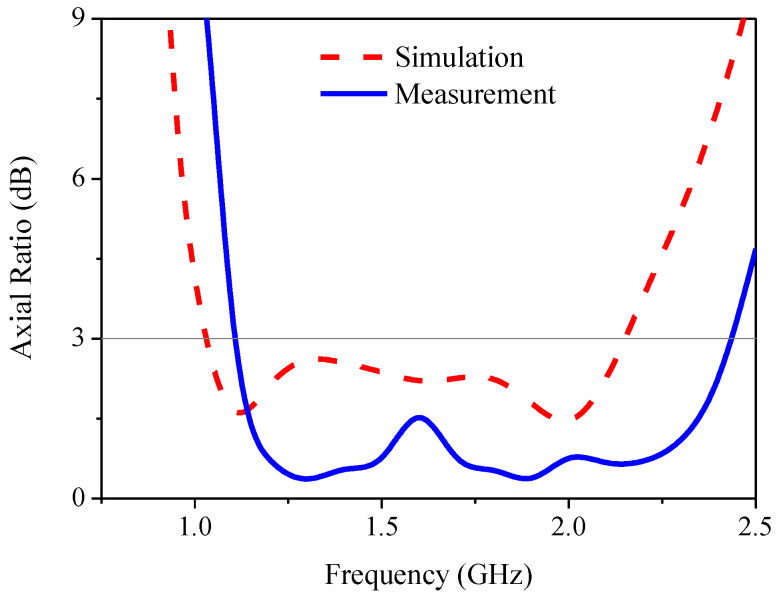
The simulated and measured axial ratio of the proposed antenna.

**Figure 20 sensors-21-01085-f020:**
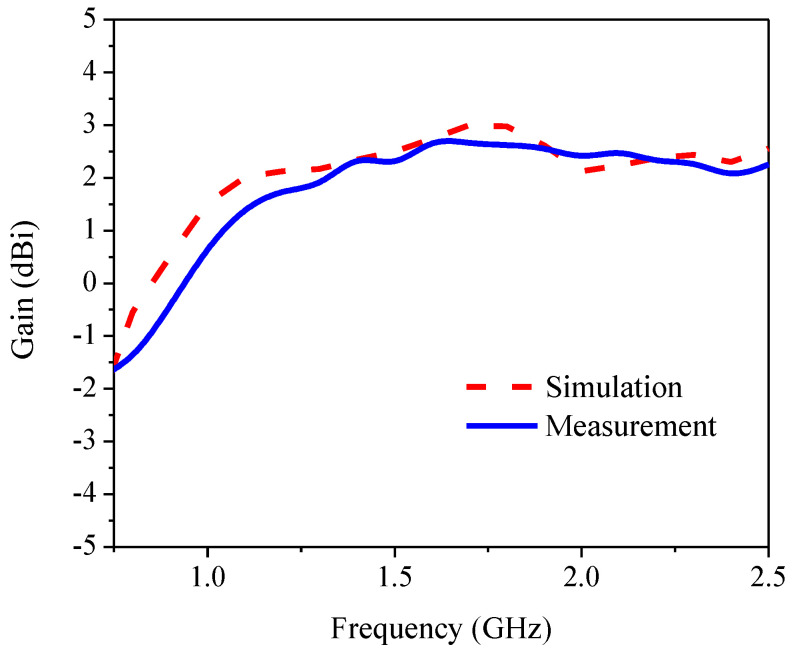
The simulated and measured gain of the proposed antenna.

**Figure 21 sensors-21-01085-f021:**
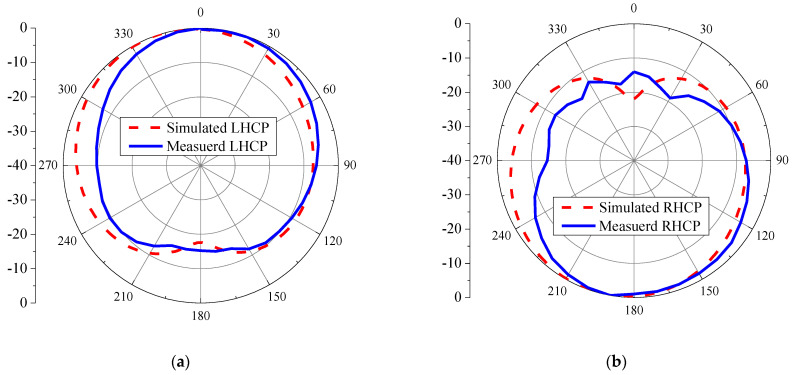
The simulated and measured radiation patterns: (**a**) LHCP, (**b**) right-hand circular polarization (RHCP).

**Table 1 sensors-21-01085-t001:** The optimal dimension of the inverted L-shaped patch antenna with corner-truncated partial ground plane diagonally adjoined with a square branch (in mm).

Parameters	*S* _1_	*S* _2_	*P* _1_	*P* _2_	*L* _1_	*L* _2_	*L* _3_	*L* _4_	*b*	*t*	*d*	*h*
Optimal value (mm)	92	96	45	68	76	2	12	29	26	27	19	1.6

**Table 2 sensors-21-01085-t002:** Optimal component values of an equivalent circuit model of inverted an L-shaped planar monopole antenna on a partial ground plane in vacuum and with FR-4 substrate.

Parameters	*R_s_*	*C* *_s_*	*L* *_s_*	*R_p_*	*C_p_*	*L_p_*
In vacuum (initial value)	4	5.9	1.9	46	5.9	1.9
With FR-4 substrate	42	5.9	3	23	15.5	1.1
Unit	Ω	pF	nH	Ω	pF	nH

**Table 3 sensors-21-01085-t003:** The extracted values for the relative permittivity of FR-4 (TLM140) at 1.5 GHz.

Measurements	1st	2nd	3rd	4th	5th	Average
**Extracted Values**	4.32	4.27	4.35	4.31	4.29	4.308

**Table 4 sensors-21-01085-t004:** The antenna parameters, return loss bandwidth (RLBW), and axial ratio bandwidth (ARBW) of existing studies and the current research.

Ref.	*f_c (RL)_* (GHz)	*f_c (AR)_* (GHz)	CP Band	RLBW (%)	ARBW (%)	Proposed Antenna/Technology
2017 [13]	1.565	1.51	L-band	34	19.86	An L-shaped tapered-feed radiator with a cross-slot ground plane.
2017 [15]	1.47	1.41	L-band	44.9	36.87	Inverted L-shaped microstrip radiator with two L-shaped branches adjoining a partial ground plane.
2018 [16]	3.12	3.48	S-band	56	63.61	Inverted L-shaped microstrip radiator with a hook-shaped branch adjoining a partial ground plane.
2018 [22]	2.82	2.92	S-band	81.06	70.55	An S-shaped microstrip loading a multiple-circular-sector patch with a slot ground plane.
2018 [23]	4.2	4.25	C-band	76.2	72.9	Inverted L-shaped microstrip radiator with a slit ground plane and spiral stubs.
2019 [24]	2.29	2.55	S-band	45.41	30.8	A simple microstrip feedline with a circular ring slot using a pair of asymmetrical rectangular slots for the ground plane.
This work	1.55	1.78	L-band	62.37	77.87	An inverted L-shaped patch antenna with a corner-truncated partial ground plane diagonally adjoined with a square branch.

## Data Availability

The data used to support the findings of this study are available from the corresponding author upon request.

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
