# Peer review of "Inverted L-Shaped CP Patch Antenna with Corner-Truncated Partial Ground Plane Diagonally Adjoined with Square Branch for L-Band Applications"

_sensors, 2021, doi:10.3390/s21041085_

Round 1
Reviewer 1 Report
This research proposed an inverted L-shaped patch antenna with corner-runcated 277 partial ground plane diagonally adjoined with square branch for the L-band spectrum (1 - 2 GHz). The paper is well organized and proposed good work in the antenna design. I have some comments for the authors:
(1) In Fig.16, can you explain why the measured return loss within the band 1.8 GHz to 2.0 GHz is degraded, and also the simulated return loss surpasses the real measurement within the band 1.0 GHz to 1.5 GHz. There might be some issued required for the further analysis like the antenna manufacturing, your real matching and etc. The same discussion are also expected in Fig. 17.
(2) Table. 2 shows the comparison with the existing work. In my opinion, it is not so impressive to find the merits of your work from this table, especially from line 257 to 260. The authors show much work in other bands besides L band but what is your intention to show the work in other bands?
Reviewer 2 Report
The authors propose an inverted L-patch CP patch antenna for L- band applications. The paper addresses the different aspects of the patch antenna design but requires more information to make it thorough and complete. Please address the following questions/concerns before revising.
1) In the introduction, please add more application-specific information on the intended use of the proposed patch. Please move table 2 to the introduction.
2) Please include the loss tangent of FR4 and the appropriate reference for the dielectric constant.
3) Keep the figures to just L-band rather than an extended frequency range on x-axis. (Figs 3 & 4).
4) Please provide the equivalent circuit model for Fig 2a and Fig 2b and explain how the introduction of FR4 increased the resonance frequency.
5) Please explain more intuitively (perhaps with an equivalent circuit) for antennas shown in Fig 5 rather than based on CST results. This will provide a better understanding of the parametric studies for the broader audience interested in reproducing such an antenna design.
6) One of the biggest challenges is the miniaturization of such an antenna. Could the authors briefly discuss the path forward in terms of miniaturization?
7) What are the limitations of the proposed antenna? Please provide future work.
Other comments:
- Always leave a "space" between a number and its unit. For example 50 m and not 50m.
- The figures are not with a similar aspect ratio as well as similar font/style/size. Please re-do them. Say, for example, fig 9, the legend is too small.
Reviewer 3 Report
The introduction must be revised with proper references and with more elaborate data, not only ARBW. All the references should have common parts with the proposed model and should have the same frequency range.
At line 112 it’s mentioned some values for equation 1 but no justification for those is met in the text.
How Er and the thickness of the board were chosen? On the market are a lot of FR-4 types that have different values. Some of them have Er greater than 4.3 and the thickness lower than 1.6mm for example. How the authors intend to measure the Er of the FR-4 used to see the proper value?
How are chosen the form of the antenna, the length of the feed line or the feed line reference point? There must be some mathematical approach for chosen the best design. From my point-of-view the size of the antenna is too big comparing with the obtained results. For example, the standard microstrip patch antenna size at 1 GHz is 92 mm x 72 mm and the proposed model has 92 mm x 96 mm, which should offer a higher gain, which isn’t appropriate if we see the obtained results.
In Figure 5 there are some evolutionary stages of the proposed antenna model, but no details are given why those specific geometric models are chosen.
For comparison should be considered the same type of antennas, in the same band. You cannot compare an L-band antenna type with an S-band antenna (the frequency range is different). The table must be explained, why those parameters are important to be mentioned and why is the best the solution from this paper.
In the text is mentioned that some measurements are performed but no images that sustain that affirmation are in. The author should give measurements details in the anechoic chamber with the proposed prototype.
Minor observations:
There must be a space between the value and the measurement unit – line 92 for example.
The template must be revised with the journal template.
Round 2
Reviewer 2 Report
Thank you for revising. I am satisfied with the responses.
Author Response
Thank you very much to satisfy our revisions.
Reviewer 3 Report
The manuscript was improved accordingly with the requirements from the comments. However, some remarks cannot be take into account due to their non-scientific and unprofessional explanation.
- According to the public datasheet of the substrate, the values indicated aren't the one from the datasheet (thickness and relative permittivity), so from my POV those values were chosen arbitrary, as generic values known from forums or other informing sources, without any scientific knowledge background. For proof of the chosen values, the authors should provide some measurement data performed on that substrate type (line 98). The question from the previous review isn't solved so, "How the authors intend to measure the Er of the FR4 used to see the proper value?". The authors should add also as reference the datasheet of the used material for the substrate.
- The mathematical approach isn't possible using only one generic equation. The authors introduced Figure 5 and Table 2 with some empirical values, having no background on those depicted values. Please provide more information on how the structure is chosen and how values are mathematically obtained.
